# Unveiling Recent Trends in Biomedical Artificial Intelligence Research: Analysis of Top-Cited Papers

Benjamin S. Glicksberg [1,2,*] and Eyal Klang [2,3,*]

1   Hasso Plattner Institute for Digital Health at Mount Sinai, Icahn School of Medicine at Mount Sinai, New York, NY 10029-6574, USA
2   The Division of Data Driven and Digital Medicine (D3M), Icahn School of Medicine at Mount Sinai, New York, NY 10029-6574, USA
3   The Sami Sagol AI Hub, ARC Innovation Center, Sheba Medical Center Affiliated to Tel-Aviv University, Tel Aviv 6139001, Israel
*   Correspondence: benjamin.glicksberg@mssm.edu (B.S.G.); eyal.klang@mountsinai.org (E.K.)

**Abstract:** This review analyzes the most influential artificial intelligence (AI) studies in health and life sciences from the past three years, delineating the evolving role of AI in these fields. We identified and analyzed the top 50 cited articles on AI in biomedicine, revealing significant trends and thematic categorizations, including Drug Development, Real-World Clinical Implementation, and Ethical and Regulatory Aspects, among others. Our findings highlight a predominant focus on AIs application in clinical settings, particularly in diagnostics, telemedicine, and medical education, accelerated by the COVID-19 pandemic. The emergence of AlphaFold marked a pivotal moment in protein structure prediction, catalyzing a cascade of related research and signifying a broader shift towards AI-driven approaches in biological research. The review underscores AIs pivotal role in disease subtyping and patient stratification, facilitating a transition towards more personalized medicine strategies. Furthermore, it illustrates AIs impact on biology, particularly in parsing complex genomic and proteomic data, enhancing our capabilities to disentangle complex, interconnected molecular processes. As AI continues to permeate the health and life sciences, balancing its rapid technological advancements with ethical stewardship and regulatory vigilance will be crucial for its sustainable and effective integration into healthcare and research.

**Keywords:** AI; machine learning; multiomics; medical imaging; personal medicine; health informatics; drug development; biomedicine

## 1. Introduction

Recent artificial intelligence (AI) innovations such as transformer models based on attention mechanisms have initiated a significant shift in health and life sciences research. The impact of these models has entered a variety of domains and data types, and the development of Large Language Models (LLMs) such as ChatGPT can process multi-modal and multi-omics data simultaneously. Enhancements in AIs reasoning capabilities may play a crucial role in scientific discovery and application in healthcare workflows. The depth and breadth of AI applications in the field of biomedicine within the past decade have been extensive and varied. There are several useful references to orient readers to the background and context of AI in biology [1,2] and medicine [3,4] overall.

This paper seeks to investigate how important recent advancements (e.g., LLMs) have affected the field of biomedicine by reviewing and synthesizing the most influential AI studies in the past three years. Our focus is on identifying those groundbreaking technologies and how they have impacted the field of biomedicine. We aim to identify highly-cited key studies and categorize them into overarching categories. Furthermore, we hope to expand these insights into prevailing trends in AI applications and project potential future directions of research and development in this rapidly evolving field.

This review paper is organized as follows: In Section 2, we describe the methods utilized to perform this study, ranging from data retrieval to article selection; Section 3 provides the overall results of the investigation; Section 4 delves into a discussion of the overall trends of the top 50 papers analyzed, separated by thematic subsections; and finally, Section 5 concludes this work with a distillation of the findings in the context of the biomedical field looking forward.

## 2. Methods

An overall schematic figure is depicted in Figure 1.

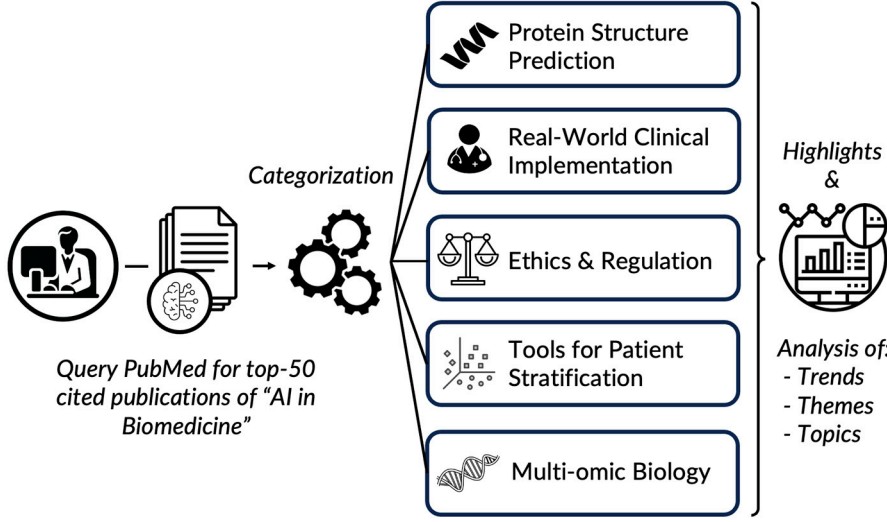

**Figure 1.** Study Overview Schematic.

### 2.1. Data Retrieval and Processing

Data for this study was retrieved on 15 November 2023, using Python 3.10. We used pymed, a Python library interfacing with PubMed, a free search engine accessing primarily the MEDLINE database of references and abstracts on life sciences and biomedical topics. Our search targeted articles related to Artificial Intelligence (AI) in life and health sciences.

### 2.2. Search Strategy

The search incorporated a combination of Medical Subject Headings (MeSH Terms) and keywords in titles and abstracts. The terms included "Artificial Intelligence", "Machine Learning", "Deep Learning", "Computer Neural Networks", "Natural Language Processing", "Computer Vision", "Reinforcement Learning", "Large Language Models", "Transformer", "OpenAI", and "ChatGPT". These terms were concatenated using the 'OR' operator to broaden the search scope, capturing a comprehensive range of articles relevant to AI advancements in life sciences.

### 2.3. Citation Analysis

We used "pmidcite," a tool for retrieving citation counts and relative citation scores from PubMed [5]. The relative citation score is a metric that compares an article's citation count to the average citation count of other articles published in the same year, providing a normalized measure of an article's impact. Self-citations were not controlled for.

### 2.4. Article Selection

From the retrieved dataset, we selected the top 50 cited articles for analysis. This selection was based on the citation counts and relative citation scores. The timeframe of the query included papers published from the beginning of March 2020 to the end of March 2023. The authors performed a manual review to ensure applicability, i.e., that the papers involved AI in health and life sciences. Some papers on the list were removed and replaced

immediately after their ranking. For instance, papers were removed if they referenced the terms (e.g., AI) in the abstract but did not actually utilize those techniques.

PubMed was queried for the top 50 cited papers in the past few years in the field of biomedicine. All papers were categorized into five common themes, and the resulting themes were analyzed and highlighted.

## 3. Results

We identified the top 50 most cited papers from a full list of nearly 75,000 works within the inspected period. All data relating to the selected papers is available in the Supplementary Materials. Based on the top 50 papers that were returned, we identified five common themes that encompass broad sets of categories and placed each paper into them (Table 1). The first category, Advancements in Protein Structure Prediction, had nine papers that focused on AI for predicting protein structures, an area that has seen significant growth and innovation, particularly in the understanding of protein folding and function [6–14]. The second category, Real-World Clinical Implementation, had 14 papers and generally explored the practical applications of AI in healthcare, including diagnostics, educational tools, and telemedicine, highlighting how AI technologies are transforming patient care and medical education [15–28]. The third category, Ethical and Regulatory Aspects, had five papers and primarily dealt with the critical examination of AI, focusing on ethical considerations, transparency, and the explainability of clinical AI systems [29–33]. The next category, Tools for Disease Subtyping and Drug Development in Personalized Medicine, had 15 papers and focused on leveraging AI tools for reclassification of disease through subtyping and patient stratification as well as how drugs can be developed towards this personalization of medicine [34–48]. The last category, Molecular and Multi-Omic Biology, had seven papers and showcased the use of AI on genomic, transcriptomic, and proteomic data for understanding complex molecular biology processes [49–55].

**Table 1.** Categorical Overview of Trends by Classification Category.

| Categories | Number of Papers | Notable Journals | Notable Paper (PMID) | Data Types | Description |
|---|---|---|---|---|---|
| Protein Structure Prediction | 9 | Nature, Science | 34265844 | Structural Data, Protein Sequences | Advances in predicting protein structures using AI are significantly impacting biological research |
| Real-World Clinical Implementation | 14 | PLOS: Digital Health, Nature Medicine | 33629156 | Imaging Data, and Clinical Data | Development and deployment of AI for diagnostic accuracy and efficiency in clinical settings |
| Ethical and Regulatory Aspects | 5 | The Lancet: Digital Health, Nature Medicine | 34711379 | Policy and Regulation Documents, Clinical Data | Highlighting the need for ethical guidelines, regulatory standards, and possibly explainability in AI applications in healthcare |
| Tools for Disease Subtyping and Classification | 15 | Nucleic Acids Research, Nature Methods | 33649564 | Imaging Data, Genomic Data | Tools and methods for how AI can be used to refine disease classification, subtyping, and drug discovery |
| Molecular and Multi-Omic Biology | 7 | Nature Genetics, Bioinformatics | 33876751 | Clinical Data, Multi-Omic Data | Advanced AI applications for understanding complex multi-omic and molecular interactions |

A summary table outlining common classification categories for the papers represented in this work. This table reflects examples of journals, notable papers, data types, and a description of each highlighted category.

## 4. Discussion

### 4.1. Highlighted Impressions and Overall Themes

In this section, we present an outline of the general themes of these highly cited works, highlighting key trends and topics as they relate to AI and machine learning in the health and life sciences. With this rapidly moving field, this examination will hopefully serve to explore how cutting-edge computational techniques are shaping recent trends in focus areas.

#### 4.1.1. Advancements in Protein Structure Prediction

Advancements in the field of protein structure prediction mark a significant leap in our understanding of biological processes and have profound implications for drug development and disease treatment. The papers in this category address the longstanding challenge of accurately predicting protein structures, a problem that has been a bottleneck in structural biology for decades. The advancements in protein structure prediction, as highlighted in the selected papers, address a critical and long-standing challenge in the field of structural biology: the accurate and efficient determination of protein structures. Before the advent of these new methods, understanding protein structures largely depended on experimental techniques like X-ray crystallography and NMR spectroscopy. These methods, while powerful, are time-consuming, resource-intensive, and often limited in their ability to resolve complex or dynamic protein structures. This limitation posed a significant barrier to a rapid and comprehensive understanding of protein functions and interactions and their implications for health and disease.

The breakthroughs represented by these papers, particularly the development of AlphaFold by DeepMind, mark a paradigm shift. AlphaFold's success in protein structure prediction using deep learning algorithms has achieved a level of accuracy that is unprecedented [7]. This method stands out because it can predict protein structures quickly and with a high degree of precision, overcoming the limitations of traditional experimental approaches. This is especially critical for proteins that have eluded structure determination through experimental means, opening new possibilities for exploring the vast array of proteins critical to understanding biological processes and disease mechanisms.

Researchers have built on AlphaFold's momentum, exploring network architectures that further refine the accuracy and efficiency of protein structure predictions [9]. Specifically, they developed a three-track neural network that can better encode information at the 1-D sequence level, the 2-D distance map level, and the 3-D coordinate level, which also allows for more rapid and accurate protein-protein complex models. Another group utilized AlphaFold for protein structure prediction for the human proteome, which resulted in a minimum increase of 19% of total residues identified, specifically from 17% to 36% [10]. To address the lingering challenge of structure prediction for proteins lacking homologous templates, another group used deep neural networks to create C-I-TASSER, which correctly folded 50% (4162/8266) domain families of unsolved Pfam families [11]. Another issue in the protein folding space is the loss of contrast at high frequencies for cryo-EM, used for protein structure modeling, which creates problems in utilization. Another group created DeepEMhancer, a deep learning solution for cryo-EM volume post-processing that was able to reduce noise levels and improve the quality of experimental maps [8].

These studies underscore the shift from traditional, experiment-based approaches to AI-driven predictions. They highlight the role of advanced algorithms in deciphering protein interactions and conformations, essential for understanding cellular processes at a molecular level. This transition to AI-centric methods is a crucial advancement, as it not only speeds up the process of structure determination but also enhances the ability to predict the interactions and functions of proteins whose structures are not easily determined through experimental means.

### 4.1.2. Real-World Clinical Implementation, including Diagnostics, Education, and Technology

A well-represented topic in this collection is the application of AI to various aspects of healthcare, particularly within the realms of predictive models, diagnostics, medical education, and technology, including telemedicine and devices. The utility of AI in clinic implementation has been extensively applied throughout the past decade. However, novel methodologies, including large language models, better system architecture, and new regulatory procedures, have decisively created a shift into more real-world applications.

The use of AI as a prognostic or diagnostic screening tool is not something new, and there are many studies showing its utility in fields like radiology, pathology, and beyond. The COVID-19 pandemic, however, revealed the potential of these tools in challenging situations. During the pandemic, health systems were unable to operate at maximum efficiency due to issues like overcrowding and resource limitations. Therefore, having systems to help synthesize and analyze data to output initial diagnoses has shown promise to help clinical practitioners make more rapid yet informed diagnoses, particularly for COVID-19. One study in this list presented an early AI algorithm that analyzed CT images for COVID-19 diagnosis and achieved a robust performance of 89.5% accuracy with 0.88 specificity and 0.87 sensitivity on internal testing and 79.3% accuracy with 0.83 specificity and 0.67 sensitivity when tested on an external dataset [20]. This performance hinted at the potential for offering a rapid, reliable alternative to traditional pathogen detection methods. Another study further advanced this domain by developing a deep learning-based CT diagnosis system that not only distinguishes COVID-19 from bacterial pneumonia with high accuracy but also identifies critical lesion features, enhancing diagnostic precision [22]. The third study employed a DenseNet201-based deep transfer learning model to classify COVID-19 cases using chest CT images [17]. This approach demonstrated superior performance over methods that existed at the time of the publication of their research. Together, these studies underscore the possibilities of AI-enabled diagnostics in times of pandemics.

Another recently popular trend in the field of biomedicine is the utility of telemedicine, which represents remote care utilizing technology like video conferencing and chatbots. Telemedicine offers accessible and efficient medical services, especially in remote or underserved areas, utilizing observations made from patient images or videos taken from a smartphone. Telemedicine can also incorporate AI analysis of such remotely taken images for initial screening and diagnostics, which can be used to determine if an in-person visit is recommended. One such work represented in this list explores the power of telemedicine within the field of ophthalmology on a global scale [18]. This review illustrates how digital innovations can enhance remote medical services, a need that became particularly pressing during the COVID-19 pandemic. It showcases AI's role in screening eye images of patients unable to routinely visit ophthalmologists in person, overcoming geographical barriers, and making decentralized delivery of certain eye care recommendations possible, thus broadening the accessibility of specialized medical services.

The last subtopic represented in this category is the role of AI in medical education. Medical training is complex and dynamic, and it is now more rapidly changing given the swift advancement of research and online dissemination, significantly outpacing the traditional timelines of medical education and knowledge acquisition. One study evaluated how ChatGPT can aid in medical training, offering a new dimension to learning that is interactive, up-to-date, and highly accessible. This is especially vital in the rapidly evolving field of medicine, where keeping abreast of the latest information is crucial [27].

Each of these studies not only reflects AIs growing influence in healthcare but also underscores the multifaceted improvements it brings in terms of technology—from remote care to education and rapid disease diagnosis.

### 4.1.3. Ethical and Explainability Concerns in AI Applications

The integration of AI into clinical settings brings forth a host of challenges and ethical considerations. As AI models seek to influence clinical decision-making, concerns regarding

bias, errors, and the interpretability of these systems become top of mind. AI models, while powerful, can inadvertently perpetuate biases present in their training data, leading to skewed or unfair medical decisions. Errors in AI predictions, especially in high-stakes medical scenarios, can have suboptimal consequences, raising questions about liability and oversight. In cases of medical errors or misdiagnoses involving AI, determining the responsible party, whether it is the healthcare provider, the AI developer, or another entity, is a complex and unsolved problem. This complexity underscores the necessity for clear guidelines and regulations around the use of AI in healthcare.

One major concern with deep learning techniques being applied in the clinic is their "black box" nature, or the fact that the architecture of these models does not allow for easy interpretation. Unlike a regression model, in which the importance of coefficients can be more easily weighed and measured, deep learning models are much more complex, involving numerous nodes, edges, layers, and weights, making insight into how the input affects the output a bit more complicated. One paper in this list argues that explainable AI is crucial for optimal healthcare delivery and provides a survey of interpretability approaches that can be used in various application settings [29]. This paper posits that transparency, in the form of explainability and interpretability, measures reliability. The other work included in this list provides a different point of view. The authors argue that, as of now, incorporating transparency techniques into deep learning models is a "false hope" and cannot reliably mitigate bias in their applications [31]. They showcase how various failure cases can still obfuscate the clinical decision-making process, such as explainability techniques being used for justification of a decision, reducing vigilance, and hampering practitioners' ability to detect when a serious error was made for an individual patient. Finally, they argue that more comprehensive testing and external validation can be used to mitigate biases. These studies both agree that there is a need for more advanced techniques that offer greater and more accurate transparency.

The other subtopic of this category revolves around the need for regulation of AI-based medical devices. One work highlights the challenges faced by regulatory bodies in keeping pace with the rapid development of AI technologies in healthcare [30]. The authors identified 222 and 240 devices approved in the USA and Europe, respectively. The paper underscores the need for stringent regulations to ensure the efficacy, safety, and quality of AI/ML-based medical devices. Governmental regulation is particularly important given the rising number of such devices being approved for clinical use. The paper calls for more transparency in the regulatory process to bolster public trust in these advanced technologies. Put together, this topic explores the ethical and explainability issues that need to be addressed in AI for healthcare. These issues include the potential for bias and errors, the need for better transparent and interpretable AI models, and the development of robust regulatory frameworks. Addressing these concerns will be necessary for long-term success in the responsible and effective use of AI in improving patient care and outcomes.

### 4.1.4. Tools for Disease Subtyping and Drug Development in Personalized Medicine

An emerging trend in personalized medicine is the exploration of the fact that complex diseases are multi-faceted and heterogeneous in nature. It is becoming increasingly clear that disease etiologies, manifestations, and outcomes have patterns at the molecular and phenotypic level that can be differentiated between groups of patients. For true precision medicine to exist, the challenge of accurately diagnosing and treating complex diseases hinges significantly on the ability to effectively subtype these conditions. Disease subtyping is when one heterogenous disease can be stratified into subgroups that have differentiated patterns, leading to targeted and personalized treatment plans. This facet is particularly vital in cancer treatment, where the effectiveness of therapies can vary greatly depending on the specific type and stage of cancer. The outcome of using tools to address these goals can help drug development itself. One review paper describes how AI can be applied across the lifecycle of pharmaceutical products, from drug discovery to clinical trial design [35]. On the development side, the review article further describes AI applications in drug

repurposing, screening, and chemical synthesis. The paper also provides a useful table for AI tools that have been developed for processing different types of data in silico. In terms of disease applications, one study developed a deep learning methodology in computational pathology to efficiently analyze whole-slide pathology images, which are time-consuming to annotate and categorize [39]. The authors developed clustering-constrained-attention multiple-instance learning (CLAM) to identify subregions of high diagnostic value, to accurately classify whole slides, and for instance-level clustering. They applied CLAM to three tasks, specifically the subtyping of renal cell carcinoma and non-small-cell lung cancer, as well as the detection of lymph node metastasis, and showed that it was able to localize well-known morphological features. In addition to the subtyping of known cancers, like in the prior work, another paper created a deep-learning-based algorithm, specifically Tumor Origin Assessment via Deep Learning (TOAD), to classify cancers of unknown primary origin. Identifying the cancer's source is critical for determining the most effective treatment strategy [43]. This technique had strong performance, achieving a top-1 accuracy of 0.83 on an internal, held-out set and 0.80 on an external testing set.

In addition to analyzing images for disease classification and subtyping, analyzing the connection between genotypes and phenotypes is pivotal in personalized medicine as it can reveal underlying genetic and biological features that differentiate etiologies and outcomes. As such, with the increased availability of larger and richer patient datasets, performing this analysis has become computationally expensive, especially when accounting for relatedness within the population structure. As such, one group has developed REGENIE, which is designed to fit a whole-genome regression model for quantitative and binary traits in a computationally efficient manner while maintaining statistical efficiency [45]. By accommodating multiple phenotypes and requiring only local segments of the genotype matrix, REGENIE substantially reduces compute time and memory usage, which the authors demonstrate in large-scale studies utilizing the UK Biobank. This work exemplifies AI approaches that create solutions for effective analyses of personalized medicine-related datasets, which are growing at substantial rates and for which traditional techniques are infeasible. Collectively, these studies underscore the pivotal role of AI methods in overcoming the challenges of disease subtyping, enhancing diagnostic accuracy, and paving the way for large-scale genotype/phenotype analyses to discover patterns that produce treatments targeted towards individual patient genetic profiles.

### 4.1.5. AI in Molecular and Multi-Omic Biology

Molecular and multi-omic biology, including genomics, transcriptomics, proteomics, and beyond, is extraordinarily complex. Amazing progress in understanding this space has emerged in recent years, partially due to the increased capabilities of measuring these components. The resulting humongous volumes of data, however, pose challenges for analysis using conventional approaches. Therefore, AI is being used for processing, analyzing, and interpreting associations in this space. This subsection comprises recent AI methods that have been developed to unravel this biological complexity, from understanding the intricacies of genetic splicing to the subtleties of drug-target interactions and beyond. These following papers showcase how AI enhances our understanding of molecular interactions and genetic regulation but also reshapes the methods and models we use to interpret the vast and intricate data.

Understanding the multiplicative ways in which regulatory DNA elements affect gene expression in various cell and tissue types is useful for mechanistic insights into potential drivers of disease. One work tackled understanding the complex cis-regulatory code, particularly how transcription factor binding motifs are arranged and interact within the sequence of DNA. As hinted, the complexity of these interactions within the genome has made it difficult to predict and understand how specific TFs influence gene expression. The authors created a deep learning model named BPNet to predict base-resolution chromatin immunoprecipitation-nexus binding profiles of transcription factors [51]. BPNet was able to uncover the arrangement of transcription factor binding motifs. This model represents a

significant leap in understanding the complex cis-regulatory code in genomics. Another work addressed a separate challenge in a similar area. There is a longstanding challenge of accurately predicting gene expression from noncoding DNA sequences. Traditional methods have struggled to account for the influence of long-range interactions in the genome on gene expression. This paper introduces a deep learning architecture called Enformer, which significantly improves gene expression prediction by integrating information from these distant interactions up to 100 kb away [55]. The advancement enables more precise variant effect predictions on gene expression and enhances the understanding of enhancer-promoter interactions directly from DNA sequence data.

Moving to proteomics, an active challenge in the space is mapping the billions of amino acids across hundreds of millions of protein sequences to function. One group utilized unsupervised protein language modeling, specifically a deep contextual language model, on sequence information to form representations about primary, secondary, and tertiary structures of proteins and beyond [53]. The authors applied this model for predicting mutational effects, secondary structures, and long-range contact. Once association analyses have been discovered in the –omics space, another challenge is how one can assign function and insights from the multitude of findings. Gene set enrichment analysis is one such method to discern patterns of function across biological domains. However, there may be an ensuing multitude of pathways that are significant, and another challenge remains in terms of how to prioritize those. Accordingly, one group developed KOBAS-i, which reflects a development over prior iterations that now incorporates overrepresentation analysis, functional class scoring, and pathway topology scoring [54]. They use their machine learning method, CGPS, which incorporates several of the aforementioned scoring tools to create one ensemble score for more intelligent prioritization and exploratory visualization. Collectively, these works showcase A's impact on molecular and multiscale biology, offering novel insights and methods that overcome challenges in traditional approaches, such as efficiency and complexity, in understanding the intricate mechanisms of human biology.

*4.2. Overall Trends in Research*

Analyzing the categories of publications in aggregate can reveal some trends about recent highly cited research in AI within the health and life sciences. It is important to remember, however, that these trends are for the selected papers only and are not necessarily reflective of the entire field. Most articles were published in the Real-World Clinical Implementation as well as Tools for Patient Stratification categories, indicating that healthcare aspects dominate the trends (Figure 2). This trend is corroborated, as clinical data and medical imaging data dominated these two categories. As expected, most of the Real-World Clinical Implementation papers involved medical imaging data of some kind, which has become increasingly utilized in the past decade, along with Electronic Health Record research [56]. Original Research articles were similarly represented in all categories except Ethics and Regulation which may be expected as many of these works are discussions around considerations of the field.

In terms of trends over time (Figure 3), a clear inflection point emerged for the Protein Structure Prediction category with the introduction of AlphaFold in July 2021. This monumental release seeded major future publications on various aspects of understanding protein structure using the methodology that explained the surge of work soon after in that category. The Real-World Clinical Implementation papers were the most steadfast across time and had early representation, with pivotal works around medical image analysis arising during the COVID-19 pandemic. The use of AI in multiomics also occurred around this time in other areas in addition to protein structure, including genomics and transcriptomics, as seen with the rise of publications in this space in early 2021. The developing ways to characterize multi-omics and medical images (seen in Clinical Implementation) also facilitated the rise of Tools for Disease Subtyping and redefining disease classification. These works also fed into the drug development field and contributed to the rapid growth

of this space at the end of 2021. As more and more multi-omic and multi-modal patient data becomes available, along with the advancement of technology and machine learning architectures, the utility of AI in biomedicine will continue to grow with proper safeguards in place from regulation.

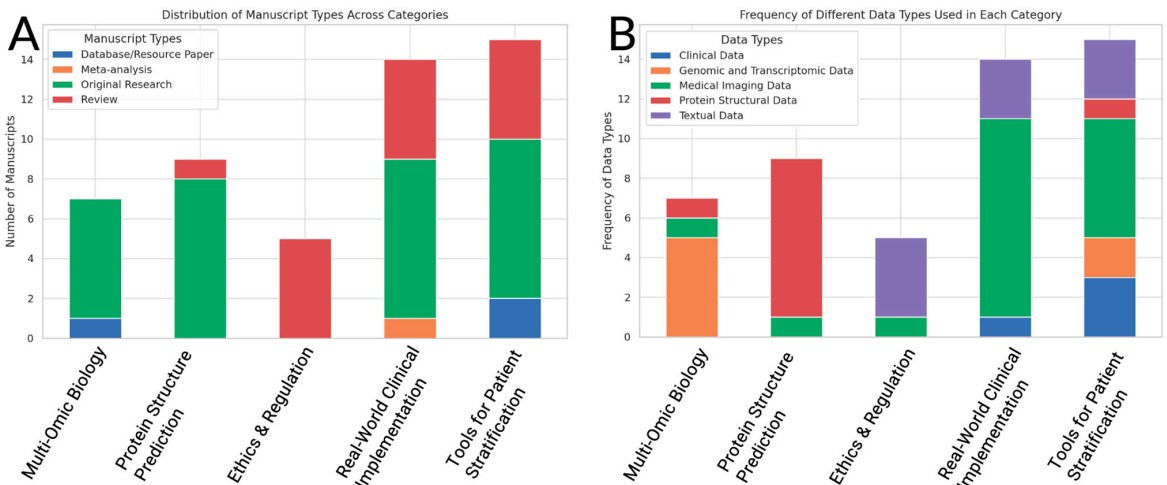

**Figure 2.** Overview of Manuscript Types and Data Types by Category of the Top 50 Papers. (**A**) Distribution of manuscript types by category. Original research was the most represented manuscript type, followed by reviews. (**B**) Distribution of data types by category. Medical imaging data was the most represented, and several categories encompassed multiple types, like Tools for Patient Stratification. All counts reflect only the top 50 papers included and are not reflective of the field overall.

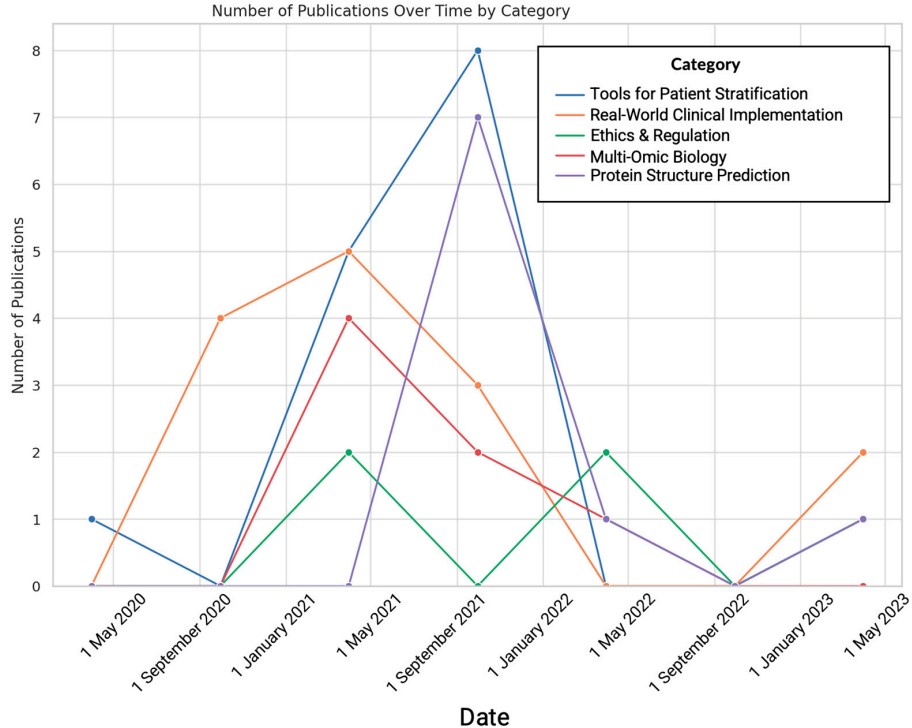

**Figure 3.** Trends in the top 50 publication numbers by category over time. This plot shows the number of publications by category over the past 3 years in half-year increments. These numbers reflect only the selected top 50 papers and should be considered within that context.

## 5. Conclusions

This paper's exploration of the top 50 cited AI studies in health and life sciences over the past three years showcases rapid transformation, driven by key groundbreaking advancements. The surge in AI applications spans a broad spectrum in biology, from revolutionizing protein structure prediction with tools like AlphaFold, which marked a paradigm shift in structural biology, to genomics and transcriptomic profiling. There is also early permeation into clinical practices with computer-aided diagnostics, telemedicine, and medical education. The COVID-19 pandemic facilitated many AI studies that were developed and employed in a much more rapid timeframe than before. At the same time, the continued incorporation of AI into healthcare has brought to the forefront critical ethical and regulatory challenges. Issues surrounding the explainability of AI systems, potential biases in AI-driven decisions, and the necessity for robust regulatory frameworks highlight the complexity of responsibly integrating AI into high-stakes medical scenarios. The ethical considerations and the push for transparent AI underscore the need for a balanced approach that prioritizes patient safety and ethical standards.

There are several limitations to this work. First and foremost, the selection criteria that we used for the citation count do not necessarily denote impact or importance. Additionally, the query approach was imperfect, included incorrect papers (which were curated), and may have missed relevant ones. Additionally, the categories and themes identified do not necessarily reflect the grand scope of work in this space and are restricted to a small subset of related published research. Accordingly, any insights derived should be held within that context. Lastly, any temporal analysis does not consider important factors like time to publication, which may vary by subfield and journal.

Furthermore, the evolution of AI in disease subtyping and drug development signifies a move towards more personalized medicine, where tools are instrumental in tailoring treatments to individual genetic profiles. This shift towards precision medicine is mirrored in the increasing focus on molecular and multi-omic biology, where AI is being leveraged to untangle complex biological data, offering unprecedented insights into genetic regulation and molecular interactions. The future trajectory of AI in this field is poised for continued innovation and expansion. However, many safety standards and rules are still being written, and careful regulation still needs to be developed to oversee implementation into practice. As AI continues to evolve and integrate deeper into the health sciences, it holds the promise of not only advancing our technological capabilities but also enriching our understanding of the complexities of human life and health.

**Supplementary Materials:** The following supporting information can be downloaded at: https://www.mdpi.com/article/10.3390/app14020785/s1, Supplementary Table S1: publication and categorization information relating to the top 50 manuscripts analyzed.

**Author Contributions:** Conceptualization; methodology; formal analysis; data curation; writing—original draft preparation.; writing—review and editing: B.S.G. and E.K. All authors have read and agreed to the published version of the manuscript.

**Funding:** This research received no external funding.

**Institutional Review Board Statement:** Not applicable.

**Informed Consent Statement:** Not applicable.

**Data Availability Statement:** Data relating to the selected manuscripts can be found in the Supplementary Materials.

**Conflicts of Interest:** The authors declare no conflict of interest.

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
