# Peer review of "Unveiling Recent Trends in Biomedical Artificial Intelligence Research: Analysis of Top-Cited Papers"

_applsci, doi:10.3390/app14020785_

Round 1

Reviewer 1 Report

Comments and Suggestions for Authors

Unveiling Recent Trends in Biomedical Artificial Intelligence Research: Analysis of Top-Cited Papers

This narrative literature review identifies the top 50 most-cited articles on artificial intelligence (AI) in biomedicine and proposes an overview of the 5 identified directions in which highly cited research has been published in the field of biomedical AI over the past 3 years.

The review is well-presented, in a logical sequence and a well-structured form, with relevant illustrations. It offers a welcome synthesis of recent AI developments in the biomedical field.

The presentation clarity and use of English language are excellent, with only few changes that should be performed for improved clarity, as suggested below:

Page 3 – in Table 1 consider replacing ”#“ with “Number of Papers”;

Page 3 – in Table 1 consider adding “PMID” after ”Notable Paper“;

Page 5, lines 169-170  – consider replacing

“[…] and achieved robust performance, specifically 89.5% accuracy and 0.88 specificity and 0.87 sensitivity on internal testing and 79.3% accuracy with 0.83 specificity and 0.67 sensitivity.”

with

“[…] and achieved a robust performance of 89.5% accuracy, with 0.88 specificity and 0.87 sensitivity on internal testing and 79.3% accuracy with 0.83 specificity and 0.67 sensitivity when tested on an external dataset.”;

Page 5, line 172 – consider replacing “pathogenic tests” with “pathogen detection methods”.

Reviewer 2 Report

Comments and Suggestions for Authors

Dear Authors,

I thought this was a very interesting idea for a paper and offered some good insights into how AI is being used in the field.

In general, it would be helpful if you could add more historical context and general context to this work. How many AI related papers did you find in your search?  What were the exact time frames you looked and is this different than previous years? Several times you make reference to a ‘surge’ in papers but without any context its difficult for the reader to understand, how much of a surge, is it a significant change from the past?

In the citation analysis section, I would like to understand the number of citations for each of these papers. Did you control for self citation in any way? Are they from mostly the same groups? This information is not found in the supp material either.

In the article selection section, you say that some articles were removed as inapplicable. What was wrong with them, how many were removed? What does this say about your search methods?

The first paragraph of page 6 is very vague. Please add details about what exactly they are doing with AI? You list some things are sound more like use of telemedicine but do not explain what AI is doing in these contexts.

‘Enhance remote medical services’, how?

‘over coming geographical barriers’, how?

How is AI delivering eye care?

Overall trends section:

I do not think that you can really make very broad ascertains about the trends from just 50 papers. It is interesting but if you want to make overarching statements, you should pull in numbers of all the papers you searched, not just 50.

In the second half of page 9, there is a feeling that AI in clinical medicine is very new, in the past three years of your search. AI and ML have been used in radiology and clinical medicine long before covid.  I feel like many of these statements need more historical context.

The do not think that figure 3 (again with only the 50 papers) is very useful, and could be misleading given the time that it takes to get paper published in different journals.

Some sentences could be reworded for clarity:

Line180-181 is confusing in general and what is ‘and beyond’?

Line 222-223, please provide some more detail or an example of what you are talking about.

Line 290-291, this sentence is very awkward, consider rewording.

Line 323, spacing on the citation is off.

Comments on the Quality of English Language

Very well written overall. Only a few sentences that need work, listed above. 

Reviewer 3 Report

Comments and Suggestions for Authors In the opinion of this reviewer, this review work, before being published, should modify or include the following: 1) Its structure (Sections, etc) must be included at the end of the Introduction. 2) In Table I, the symbol # must be defined. 3) Section 5 of the manuscript (AI in Molecular and Multi-Omic Biology)... Is it a Subsection of 4. Discussion? 4) Overall Trends in Research…Is it a Section or a Subsection?

Round 2

Reviewer 3 Report

Comments and Suggestions for Authors

The new version of the manuscript is correct.